# Brief Overview on Bio-Based Adhesives and Sealants

**DOI:** 10.3390/polym11101685

**Published:** 2019-10-15

**Authors:** Solange Magalhães, Luís Alves, Bruno Medronho, Ana C. Fonseca, Anabela Romano, Jorge F.J. Coelho, Magnus Norgren

**Affiliations:** 1CIEPQPF, Department of Chemical Engineering, University of Coimbra, Pólo II—R. Silvio Lima, 3030-790 Coimbra, Portugal; luisalves@ci.uc.pt; 2Faculty of Sciences and Technology (MeditBio), Ed. 8, University of Algarve, Campus de Gambelas, 8005-139 Faro, Portugal; aromano@ualg.pt; 3FSCN, Surface and Colloid Engineering, Mid Sweden University, SE-851 70 Sundsvall, Sweden; Magnus.Norgren@miun.se; 4CEMMPRE, Department of Chemical Engineering, University of Coimbra, Rua Sílvio Lima-Pólo II, 3030-290 Coimbra, Portugal; ana.clo.fonseca@gmail.com (A.C.F.);

**Keywords:** cellulose, lignin, adhesives, sealants, silicone, adhesion

## Abstract

Adhesives and sealants (AS) are materials with excellent properties, versatility, and simple curing mechanisms, being widely used in different areas ranging from the construction to the medical sectors. Due to the fast-growing demand for petroleum-based products and the consequent negative environmental impact, there is an increasing need to develop novel and more sustainable sources to obtain raw materials (monomers). This reality is particularly relevant for AS industries, which are generally dependent on non-sustainable fossil raw materials. In this respect, biopolymers, such as cellulose, starch, lignin, or proteins, emerge as important alternatives. Nevertheless, substantial improvements and developments are still required in order to simplify the synthetic routes, as well as to improve the biopolymer stability and performance of these new bio-based AS formulations. This environmentally friendly strategy will hopefully lead to the future partial or even total replacement of non-renewable petroleum-based feedstock. In this brief overview, the general features of typical AS are reviewed and critically discussed regarding their drawbacks and advantages. Moreover, the challenges faced by novel and more ecological alternatives, in particular lignocellulose-based solutions, are highlighted.

## 1. Background

In the construction area different types of adhesives and sealants (AS) are used, such as polyurethane (PU), epoxies (EP), polyamides (PA), ethylene-vinyl acetate-copolymers (EVA), poly(vinyl acetate)s (PVAc), silicones, etc. It is estimated that more than 20% of the AS produced worldwide are applied in the construction field, namely in window and door profiles, pipes and guttering, flooring, glazing, insulation, building panels, and roofing [1]. In this respect, wood adhesives emerge as a very important class considering the huge amount consumed worldwide [2,3,4]. AS also find application in the electronics, automotive, and energy sectors [5]. Among the different AS those based on silicone stand out due to their superior features, such as elasticity, versatility, good adhesion, durability, and UV resistance [5,6,7].

Globally, the annual amount of manufactured silicone-based AS products is ca. 2,122,000 metric tons with an estimated revenue of more than €10.6 billion [8]. Despite being clearly a profitable sector, silicones are quite expensive and non-environmentally friendly, since most of the silicones used nowadays are obtained from non-sustainable and non-renewable fossil-based feedstock and present a very limited biodegradability rate [9,10], mainly due to their very low water solubility [11,12,13,14].

The fate of silicones in soil is largely unknown but some studies have suggested that in a clay loam, polydimethylsiloxane (PDMS) degrades quite slowly and after six months it yields only ca. 3% of low-molecular-weight water soluble products [11]. This slow degradation rate can be related to the high humidity in the soil, since low moisture levels are beneficial for a more efficient and faster degradation [11,12,13,14].

Lately there has been an increasing silicone shortage, particularly in the building sector, due to its more recent uses silicone use in the medical area. Strict environmental regulations have also been affecting the sector driving a marked growth in consciousness to reduce the fossil dependence and mitigate the global pollution [15]. Consequently, the demand for products manufactured from renewable and sustainable sources, such as biopolymers, has significantly increased, resulting in remarkable scientific and industrial advances [16,17]. These advances can lead to a decrease of manufacturing costs while creating new whole “green” businesses opportunities. The more interested reader is guided to the excellent recent review of Heinrich, which discusses the future challenges and opportunities of bio-based adhesives [18]. The use of biopolymers in the construction sector is already quite significant mainly due to poly(lactic acid) (PLA) and polyhydroxyalkanoates (PHA), which have been driving the fast growth of the bio-based polymer industry in the recent years [19,20,21]. PHA has also been exploited as bioplastics, fine chemicals, implant biomaterials, medicines, and biofuels. Additionally, PHA have been suggested to improve the robustness of industrial microorganisms and contribute to the regulation of bacterial metabolism [22]. The applications of PLA are mainly as thermoformed products, such as drink cups, take-away food trays, containers, and planter boxes [23]. Despite the significant progress and potential, the biopolymers presently used in AS formulations, such as cellulose derivatives, carrageenans, and agarose, are still not fully suitable to replace silicone-based ones mainly due to limitations on their performance [18].

## 2. Organosilicon Based Adhesives and Sealants

As mentioned above, silicones are the most commonly used AS [24]. These consist of organosilicon polymers whose backbone is mainly composed of Si–O and Si–C linkages (Figure 1) [25]. Organosilicates are silicates in which Si–O is bonded to an organic group or even to a more complex organic structure. Such structures are considered to be inorganic because the silicon atom is bonded to carbon through an oxygen atom unlike the organosilicon structures, where the silicon atom bonds directly to carbon. The organic moiety is typically a methyl or an ethyl group [3]. The most common example is PDMS, a synthetic polymer with a (CH_3_)_2_SiO repeating unit.

Silicones display the unusual combination of an inorganic chain (similar to silicates and often associated with high surface energy) with organic methyl side groups that are often associated with low surface energy [26]. Also, silicones are characterized by high flexibility, a feature that stems from the relatively long Si-O and Si-C bond (Si–O ca. 1.64 Å and Si–C ca. 1.88 Å). This allows the silicone molecule to adopt many configurations (high configurational entropy) [27]. For instance, while the rotational energy of the CH_2_–CH_2_ bond in polyethylene is ca. 13.8 kJ/mol, the rotational energy for the Me_2_Si–O bond is ca. 3.3 kJ/mol, which can be regarded as “free” rotation [28].

Organosilicon-based polymers containing inorganic structural elements typically display some remarkable properties, such as excellent resistance to chemicals and adverse weather conditions and remarkable adhesion to silicate materials (e.g., ceramics, enamel, glass, concrete, and others) and metals (especially aluminum). Moreover, these systems often present high heat resistance and flexibility at low temperatures, good elastic recovery (resilience), and low shrinkage during crosslinking [6].

Typically, the crosslinking reaction of silanes consists of two steps, hydrolysis and condensation, as schematically presented in Figure 2 and Figure 3, respectively. When stored under inert gas (N_2_), the silane monomers are non-reactive, in the form of FG–Si–OR where –R is the alkyl group and FG represents the organic functional group. However, alkyltriacetoxy silanes can be hydrolyzed by moisture, originating the –Si–OH moiety. This group is also known as “silanol” and can readily react to substrates or fillers containing OH groups [29].

After the hydrolysis of alkyltriacetoxy silanes the condensation process may take place. This results in the adhesion of silanes to the substrates or coupling/dispersing agents to the fillers, as schematically illustrated in Figure 3.

## 3. Fundamentals of Adhesion

The main difference between sealants and adhesives is that the former typically presents lower strength and higher elongation in comparison to adhesives. Sealants are often used to seal joints between building components or materials [29]. Adhesives can be obtained as one-component products containing all the reactive compounds in a unique mixture, and two-component products in which the reactive substances are mixed immediately before the product application. Joints often contain a polymeric substance that is linked to the substrate via chemical bonding, physiochemical attractions, and physical interactions. The way the polymer is applied is as important as its chemical composition, since it determines the conditions under which the application must take place and therefore the possible end-uses. It also affects other factors, such as the spreading of the adhesive on a certain matrix and the area of contact, which in turn have a substantial impact on the resulting adhesive forces [18].

Adhesives often offer a more rigid and durable performance, since they are typically used to join two substrates. Despite the differences, AS materials share many features including critical aspects related to their adhesion capacity. In order to hold surfaces together, fluid AS are preferred to be applied to a certain substrate in order to wet, spread, and completely penetrate the surface leaving no empty voids. Hence, AS of low viscosity are preferred at the time of application. Additionally, in order to provide strong cohesive strength, the AS must be set or solidified by either cooling, crosslinking reactions, or via solvent evaporation, depending on whether the adhesive is hot-melt (thermoplastic), thermoset, or solvent-based [18,31].

The adhesion strength is a measure of the resistance of an adhesive bond to the mechanical removal from a substrate. It is represented as force/area (MPa or N.mm^−2^) or force/length and commonly named “peel strength” (N.mm^−1^) [24]. A prerequisite for good adhesion is related to the correct hydrophobic/hydrophilic balance between the substrate and the AS [32].

The work of adhesion of a liquid to a solid, W_A_, can be described by the Young–Dupré equation (Equation (1)). It is composed by two measurable parameters; the surface tension of the liquid and the contact angle between the liquid and the surface [33,34].
(1)$$W_{A}=\gamma_{LG}(1+\cos\theta ).$$

A liquid, such as an AS, cannot wet the surface of a solid substrate unless its surface tension is lower than that of the substrate ($\gamma_{LG}<\gamma_{S}$). Thus, if a liquid has a higher surface tension than the substrate, wetting is not favored [35]. Depending on the substrate different adhesion mechanisms can be observed. For example, metal surfaces can interact by ionic bonds (salt formation), chelate complexes on surfaces, dipole-dipole forces, or hydrogen bonding [24]. On the other hand, plastics (containing oxidized surfaces) or wood can react chemically with the AS to form covalent bonds, through the OH groups. In this respect, silanes are considered adhesion promoters due to their high ability to react with the OH groups of the substrates. Depending on the chemical composition of the substrate different adhesion capacities can be achieved by the silane-based AS, as illustrated in Figure 4.

## 4. New Generation of Adhesives and Sealants

The growth of world population and resources consumption, combined with the concerning challenges of climate change and feedstock scarcity, will rapidly increase the demand for products manufactured from renewable and sustainable resources. Therefore, it is not surprising that the development of eco-friendly renewable products in the construction sector, such as in cladding panels, fiber reinforced polymer composite materials in bridges, and glazing sealants, has considerably increased in the last years, representing an important driving force for the continued market growth. For instance, in 2015 the global bio-based construction polymer market was estimated to ca. €10.4 billion [36]. According to the latest Grand View Research Inc. report, this number is expected to increase to nearly €30 billion in 2024 [37].

Nevertheless, so far, no entirely suitable biopolymer-based alternatives to silicones have been developed.

In what follows, a brief summary of some of the major achievements is reported mainly focusing on lignocelluloses, vegetable oils, and proteins.

## 5. Cellulose-Based Adhesives and Sealants

Cellulose is the most abundant biopolymer on the planet with an estimated annual production of ca. 1.5 × 10^12^ ton [38]. Although cellulose meets most of the requirements needed for an adhesive, due to its semicrystalline structure and complex network of interactions, such as hydrogen bonding and hydrophobic interactions, cellulose cannot be melted nor it is easily dissolved in common solvents [39,40,41]. Nevertheless, once the dissolution problem is overcome using for instance cellulose-derivatives, cellulose represents a very interesting choice in numerous applications, including as components AS agents, due to its renewability, availability, low cost, and fascinating physicochemical properties.

The first developed cellulose-based adhesive was trimethylsilylcellulose (TMSC). It is a very well-known cellulose derivative introduced by Schuyten et al. almost 70 years ago (partial structure on top of Figure 5) [42]. The TMSC was synthesized through the reaction of cellulose with different organo-chlorosilanes in the presence of pyridine. The ^13^C NMR spectrum in Figure 5 reveals the typical fingerprint of the synthesized TMSC. Only signals for the substituent (0.0–2.0 ppm) and for the anhydroglucose unit (103.0–60.8 ppm) were found. The authors have obtained TMSC with different degrees of substitution in a controlled manner, but with low solubility in some relevant organic solvents, such as in a toluene/ethanol (80/20) mixture. Later some improvements in the TMSC synthesis led to products soluble in some organic solvents, such as chloroform, 1,1,1-trichloroethane and *o*-xylene [43].

Generally, the synthesis of TMSC comprises several steps involving cellulose dissolution in a non-volatile solvent, such as N,N-dimethylacetamide with LiCl, derivatization in homogenous phase, and phase separation of the obtained TMSC. The product can be finally dissolved in a common suitable organic solvent (tetrahydrofuran or toluene) [44]. The obtained silylated cellulose is highly hydrolysable in the presence of water or another hydroxylated compound. Additionally, polymer crosslinking can be achieved as a result of either some secondary reactions of the trimethylchlorosilane itself, or due to the presence of some impurities in the chlorosilane, such as higher chlorinated silanes. The latter are known to be considerably more reactive than trimethylchlorosilane [43]. The main challenge is to control the crosslinking process, enabling the reaction of the crosslinker (chlorosilane) with the OH groups of the substrate.

Another interesting route to obtain cellulose-based compounds with potential application as AS was reported by Stiubianu et al. [47]. The authors have combined a cellulose derivative (cellulose acetate, DS ≈ 2.5) with different molar fractions of poly[dimethyl(methyl-H)siloxane]. A dehydrocoupling reaction between the Si–H and the C–OH groups occurred in the presence of the Karstedt’s catalyst, leading to the formation of Si–O–C bonds (Figure 6). The reaction proceeded at room temperature, which is quite beneficial for industrial proposes. Nevertheless, an important limitation of this procedure is related to the catalyst, which should be maintained away from cellulose acetate or other material containing OH groups due to it enhanced reactivity after being mixed with the other reagents [47].

In a related work by Klemm et al., silylated cellulose was prepared in heterogeneous phase reaction in the presence of ammonia-saturated polar aprotic solvents [48]. Multilayered supramolecular silylated cellulose structures were formed after applying a Langmuir–Blodgett technique. These ultrathin films may be also suitable for AS purposes.

Robles et al. have successfully modified the surface of cellulose nanofibrils with aminopropyl triethoxysilane (ATS) [49]. The process is suggested to occur in four steps, but it basically involves a chemical grafting on the hydroxyl groups of cellulose chains (Figure 7). The reaction was performed at 105 °C, which is not suitable for AS materials. However, some other works have demonstrated the possibility of curing of ATS at room temperature [50]. The use of ATS as a curing reagent for cellulosic raw materials appears as an interesting approach for bio-based AS, due to the possibility of chemical crosslinking between ATS and biopolymers (contained in the AS) or surfaces containing OH groups.

Farnaz Eslah et al. have studied the formation of adhesives composed of acetylated cellulose nanocrystals (ACNC), soybean flour (SF), and acetylated soybean flour (ASF) [51]. The acetylation of the cellulose nanocrystals reduces the crystallinity of cellulose. The SF reaction with acetic anhydride converts the amine and hydroxyl groups into amides and esters, respectively. Primary and secondary amines in the ASF-based adhesive formulation disappeared, which may suggest the reduction of the amine content and formation of amides. In the SF/polyethylenimine (PEI)/ACNC/NaOH formulation functional groups, such as hydroxyl, carboxyl, and amines can react and create hydrogen bonds with the hydroxyl and carbonyl groups of the ACNC. In the ASF/PEI/NaOH adhesives, the amines of the PEI reacted with the esters of the ASF to form amides. By mixing the ASF formulations with NaOH and PEI a reduction in water evaporation in the adhesives during hot-pressing was observed. These bio-based adhesives were found to fulfil the requirements for interior plywood, according to the American National Standards.

Khanjanzadeh et al. have prepared silylated CNC from the reaction of 3-aminopropyltriethoxysilane (APTES) with CNC in aqueous medium (Figure 8) [52].

The silylation of CNC was carried out through the physical adsorption, chemical grafting, and condensation reaction between −OH groups of the CNC and the silanol groups of APTES. The modified CNC were incorporated in urea-formaldehyde adhesives and their performance investigated. It was observed that the incorporation of modified-CNC improves the mechanical and other physical properties of the medium density fiberboard panels, while the formaldehyde emission significantly decreases.

Draman et al. have prepared nanocomposites of cellulose-based adhesive and polypyrrole (PPy) via a colloidal dispersion method [54]. Cellulose was chemically modified with epoxy to display adhesion properties. Different ratios of toluenesulfonic acid (TSA)-doped PPy with epoxypropyl cellulose were tested and their electrical and conductivity properties evaluated as summarized in Table 1.

Based on the data shown in Table 1, PPy and TSA-doped PPy gave electrical conductivities in the semiconductive range. The delocalization of π electrons along the polymeric backbone, co-existing with low ionization potentials and high electron affinities, led to unique electrical properties of the conjugated nanocomposites including PPy and TSA-doped PPy [55]. The authors have also shown a decrease in electrical conductivity in the cellulose-based adhesives, since cellulose works as an insulator. This non-metal material causes interface resistance, which is believed to also be the blockage for the thermal conductivity [56]. The thermal conductivity displayed suggests that this cellulose-based material can potentially be used in small electronic devices.

## 6. Starch Based Adhesives and Sealants

Starch and cellulose are two very similar polysaccharides but with different configurations of the linkages between the anhydroglucose units. While starch consists of two types of molecules amylase and amylopectin having the so called α configuration, cellulose has β configuration of the linkages forming extended ribbon-like conformations. This conformational difference not only makes starch less crystalline than cellulose but also more easily solubilized.

Biodegradable composites with high robustness and elastic properties based on corn starch and PDMS have been developed by Ceseracciu et al. [57]. The authors reported a facile method for preparing a biodegradable elastomer incorporating large amounts of unmodified corn starch (exceeding 80% by volume) in acetoxy-polyorganosiloxane thermosets. As can be observed in Figure 9, the degree of transparency of the obtained films change significantly with the amount of starch but overall the materials are quite mechanically robust. It was also demonstrated that the naturally adsorbed moisture on the starch surface enables the auto-catalytic rapid hydrolysis of the polyorganosiloxane forming Si–O–Si networks. Additionally, corn starch granules have also excellent compatibility with the addition-cure polysiloxane chemistry. Regardless of the starch concentrations used, all the developed bio-elastomers have hydrophobic surfaces with low friction coefficient and much less water uptake capacity than thermoplastic starch. The bio-elastomers are biocompatible and estimated to biodegrade rapidly even in an aquatic environment, thus avoiding one of the main drawbacks of standard silicones [57].

Sugih et al. have studied an alternative method to synthesize poly-(ε-caprolactone) grafted starch co-polymers, an interesting material to be used as compatibilizer in starch-polymer blends [58]. For each experiment, pre-dried corn starch and dimethylsulfoxide were stirred at 70 °C for about 3 h until a clear solution was formed. Then hexamethyldisilazane (HMDS) was added to the gelatinized mixture to initiate the silylation reaction. The purpose of silylation was to make starch more hydrophobic by partial substitution of the OH groups by trimethylsilyl groups. After 2–4 h reaction time, toluene was added to solubilize the precipitated and partially silylated starch. The reaction scheme is shown in Figure 10.

Note that the partially silylated starch is more hydrophobic than pristine starch and thus more soluble in organic solvents. This decrease in hydrophilicity allows a more efficient reaction with ε-caprolactone [59]. The synthesized product has been suggested as a potential compatibilizer in starch-polymer blends [58,59].

Wei et al. have used hexadecyltrimethoxysilane (HDS), a siloxane with a long hydrocarbon chain, to prepare hydrophobic modified starch nanocrystals (SNC). The use of long side hydrocarbon chain of HDS was expected to provide an efficient and simple way to improve the hydrophobicity of SNC [60]. The SNC were prepared by acid hydrolysis of waxy maize starch according to the method of Angellier et al [61]. Afterwards, the silane modified SNC was prepared by physical adsorption, chemical grafting, and condensation reactions between the hydroxyl and silanol groups, as represented in Figure 11.

As can be observed in Figure 12, the hydrophobicity and the hydrophobic stability of the modified SNC was found to increase with the HDS content.

The dispersion of modified SNC was significantly improved due to the introduction of the long chain hydrocarbon. Moreover, the hydrophobically modified SNC show great compatibility with the non-polar solvents.

## 7. Lignin Based Adhesives and Sealants

Lignin is one of the most abundant biopolymers on earth and makes up 15% to 30% of the cell walls of terrestrial plants. It is an inexpensive renewable resource which possesses numerous attractive properties, such as high thermal stability, biodegradability, high carbon content, antioxidant activity, and favorable stiffness. The use of lignin in different applications has been a topic of interest for many researchers [4,62].

Nowadays, most of the aromatic feed chemicals originate from non-renewable fossil sources. It is believed that lignin could be a very appealing future source of natural polyphenols to compete and eventually substitute petroleum-based precursors. Despite its great potential to be used in the preparation of novel materials, lignin finds limited applications. In the case of the pulp and paper industry less than 2% of the overall produced lignin is employed as concrete additives, stabilizing or dispersant agents. The majority of what remains is simply disposed or burnt as a low-grade fuel.

Stiubianu et al. have used lignin to produce an elastomer with improved properties for energy harvesting and actuation [63]. In their work, lignin additions that ranged from 0% to 50% by weight relative to siloxane do not act as stiffener for the siloxane but rather act as a bulk filler material. The systems were prepared using well-known and simple methods applied in silicone chemistry, preserving the softness and low value of Young’s modulus, characteristic of silicones. The developed elastomers have low water sorption due to their hydrophobic features and low dielectric constant. These properties were enhanced with the lignin content added to the formulation. Despite the presence of lignin, which could facilitate the conduction of electrical currents, all the samples were observed to present low conductivity, thus behaving as very interesting insulators.

Feldman et al. have developed PU materials modified with lignin [64]. The results obtained suggest that lignin acts as a reinforcing agent adding rigidity to the polymeric matrix. This was further confirmed by the increase in toughness and shore of the blended sealants. The curing mode of PU was also observed to change with the addition of lignin. Moreover, the initial setting time was reduced with the addition of lignin, but the rate of curing was found to remain constant. This suggests that the matrix is hardened in direct proportion to the amount of lignin present in the blend. It was hypothesized that the incorporation of lignin may contribute to an increase in the degree of crosslinking of the PU sealant [64].

Another interesting study was reported by Scarica et al., where a novel lignin-based thermosetting polyester (PE) coating system was developed. The material was synthesized via the functionalization of softwood lignin with succinic anhydride (SAn) (Figure 13) [65]. Materials containing different ratios of SAn/lignin were prepared and subsequently employed as macromolecular building blocks for the fully cross-linked lignin-based PE coatings (Figure 14).

This work introduces a straightforward strategy to develop new high-lignin-content PE thermosetting systems that may be suitable for bio-derived coatings and adhesive alternatives [66].

In another interesting report, lignin-based anticorrosive coatings have been developed after lignin fractionalization (LF) and silanization (LF-S) [67]. For the silanization reaction, 1-isocyanate-3-trimethoxysilylpropane (ITMSP) and dibutyltin dilaurate (DBTDL) have been used, as schematically presented in Figure 15.

The authors report four different methods to obtain lignin-based coatings. The first two methods comprise solution-based deposition of LF and LF-S, followed by appropriate thermal treatment. The third coating was obtained from a formulation containing LF-S and acetic acid, which worked as a co-crosslinking agent. Acetic acid was selected in order to determine the viability of using H^+^ as catalyst to promote the hydrolysis and condensation reactions of the methoxy groups in silanized lignin, thereby envisaging improved adhesion to the target metallic substrates (Figure 16). In the last formulation, the cross-linking agent tetraethyl orthosilicate (TEOS) was also incorporated in order to evaluate its ability to act as organic-inorganic hybrid crosslinker in the lignin-based coating system.

The latter method, where TEOS was used as a cross-linker, was found to be the most promising due to the more rigid structure originated from the presence of the inorganic phase that partially hinders macromolecular movement in the cross-linker system [67].

## 8. Other Systems: Vegetable Oils and Protein Based Silanes

Vegetable oils present versatile chemical options for different applications since they host reactive sites, such as ester groups, double bonds, and allylic hydrogens [68,69,70]. On the other hand, the use of vegetable oils as a raw material has many advantages, such as availability, price stability, sustainability, physico-chemical properties, and ecological reasons (e.g., sequestration of carbon dioxide by plants) [21]. Rapeseed is the most important oilseed in Europe and largely used in the production of biodiesel [71]. From a chemical point of view, the long aliphatic chains present in a typical vegetable oil can be used for the synthesis of for example, new silanes and polysiloxanes. These novel systems with hydrophobic features could find applications as coatings for the protection of different surfaces, such as wood, metal, or even concrete [69].

Several plants that produce unsaturated oils are claimed to be important in a healthy human diet. Their reactive unsaturated bonds also make them suitable candidates for other applications. For instance, these unsaturated oils have been used as coatings, such as the case of catalyzed oxidative curing of linseed oil finishes [72]. Another pursued application is the conversion of the unsaturated carbon bonds into polyols, which can be further transformed into PU [73]. As mentioned before, PUs have been widely used as adhesives since they only require mild curing temperature conditions, using the existing moisture of wood. Moreover, PU is more capable of bonding wood matrixes with higher moisture contents than are phenolic and amino-based resins and adhesives. Fatty oils for non-PU routes are also being investigated for making adhesives and coatings [74]. For example, vegetable oil alternatives to silanes have been developed by attaching the siloxane groups to the carboxylic groups of fatty acids from rapeseed oil [69]. Due to the problems related to glycerine removal formed during vegetable oil saponification, the process was preceded by the reaction of oil transesterification leading to methyl ester of fatty acids. The purified methyl esters were subjected to a posterior saponification and then the resultant sodium salts of fatty acids were subjected to nucleophilic substitution with 3-chloropropyltrimethoxysilane in the presence of potassium iodide as the catalyst (Figure 17).

In fact, the obtained silane following the procedure described in Figure 17 has been employed in the creation of a coating capable to protect wood against water. As introduced before, due to water exposition hydrolysis and condensation reactions may occur, leading to the formation of Si–OH groups. The subsequent two types of condensation reactions between Si–OH and Si–OH–CH_3_ moieties create a well-developed cross-linked network consisting of Si–O–Si covalent bonds. Wood-O–Si covalent bonds are then formed as a result of reactions occurring between –OH groups present in wood, and Si–OH and Si–O–CH_3_ groups present in the solution (Figure 18).

The coating is found to be attached to the surface of the modified element following the essential principle of the adhesive formulations. Furthermore, the aliphatic chain bonded to the silicon atom makes the surface of wood substrate more hydrophobic in comparison to the unmodified sample, thus forming a real protection against moisture-rich environments.

Protein-based AS have been explored particularly for biomedical applications, such as cardiac, vascular, oral, and reconstructive surgery [75]. These protein-based systems are typically used directly or in combination with a crosslinking agent that forms covalent bonds with the tissue surface [76,77]. Proteins can be directly extracted from human or other animal sources, offering many advantages over other common glues regarding biocompatibility and mechanical properties. Strausberg et al. have studied different marine organisms regarding their potential use as adhesives in classical medical and dental applications [78]. Some mussels and barnacles can produce remarkable moisture-resistant adhesives. For instance, the blue mussel *Mytilus edulis* synthesizes a specific polyphenolic adhesive protein, which plays a key role in the attachment to surfaces. The protein is located in a thread-like structure known as byssus, which also contains several other proteins including collagen. When secreted and applied, the byssal adhesive is highly cross-linked and cannot be readily analyzed. The adherence of the engineered mussel adhesive protein in an aqueous environment to various surfaces including polystyrene, glass, hydrogel, and collagen has been tested. In each case, the activation to the quinone form of the protein was required for good surface adhesion. The nature of the chemical bond involved in cross-linking has not yet been determined. However, the authors have suggested that quinone residues could bond to the c-amino group of lysine through a Michael addition reaction.

Kumar et al. have developed adhesives and plastics from soy protein. An initial protein acylation (Figure 19) was performed to improve functional properties, such as solubility and surface hydrophobicity.

It has been shown that the silanization using 3-(2-aminoethyl)-aminopropyltrimethoxy silane as a coupling agent enhances the interfacial adhesion between the soy matrix and, for instance, glass fiber to produce fiber-reinforced composites (Figure 20).

Copolymers of soy protein isolate (SPI) have also been prepared by heating an alkaline dispersion of SPI with cationic epoxide or an acrylate monomer to originate modified protein materials. The modified protein can be used alone or in combination with styrene-butadiene rubber latex for paper coating [79]. The performance of this protein-based adhesive has been shown to be dependent on the particle size, nature of the surface, protein structure, viscosity, and pH.

Liu et al. have demonstrated that although soy protein-based adhesives have relatively low strength, low water resistance, and sensitivity to biological degradation, this can be significantly improved by combining them with a functional group typically found in marine adhesive proteins. The authors have observed that this modification converts the soy protein into a robust and water-resistant adhesive with superior performance in wood applications [80]. The adhesive strengths and water resistance of plywood samples bonded with the developed bio-adhesive are competitive with commercial PF and UF resins. The modification of soy protein using marine adhesive proteins as a model, could also lead to the development of superior and water-resistant adhesives for other applications and materials than wood.

## 9. Concluding Remarks

Silicones have a major role on the adhesives and sealants market due to their superior physicochemical and mechanical properties and simple curing mechanisms. However, the poor biodegradability and recent scarce availability of silicone-based materials has been triggering the development of novel bio-based alternatives. This brief overview clearly demonstrates the opportunities introduced by renewable resources, such as cellulose, lignin, starch, vegetable oils, or proteins, in the formation of novel bio-based materials to substitute traditional adhesives and sealants with reduced human and environmental toxicity, competitive performance, and improved biodegradability.

One important factor that differentiates bio-based adhesives from traditional adhesives based on petroleum-derived materials is their varied “molecular architecture”, which can confer new functionalities to the final product. Additionally, the connections and hierarchical structures provided by nature (striking in the case of lignin and proteins) may also offer novel complex and not easily accessible features found in traditional polymerization chemistry. This versatility of many bio-based molecules may also enable their use addressing more than one function in a formulation. The use of biopolymers may also avoid the need of extensive polymerization processes, since biopolymers are already large macromolecules with a high density of functional groups. Such a high number of functional groups is expected to lead to higher crosslinking densities, which might confer superior properties in many applications.

It is also clear that several monomers derived from renewable sources often contain combinations of functional groups that would not be easily synthesized or economically viable if produced through petrochemical pathways.

The advantages of bio-based materials, and the strategies briefly highlighted in this work, pave the way and contribute for a future society less dependent on non-renewable and polluting fossil feedstock. The available examples in the literature are still scarce, partially because it is important for industries to maintain trade secrets on formulations. Nevertheless, the potential is obvious and therefore it is expected that the combination of the adhesion power of siloxane groups with the beneficial properties of bio-based raw materials will reinforce the development of novel innovative and competitive materials, in comparison to the rather non-sustainable adhesives and sealants used nowadays.

## Figures and Tables

**Figure 1 polymers-11-01685-f001:**
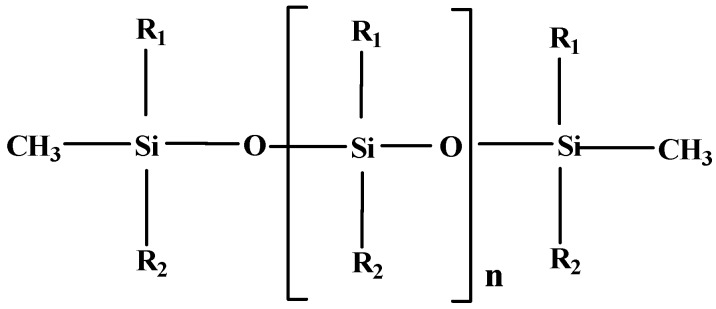
Schematic representation of a silicone molecule. R_1_ and R_2_ represent different substituent groups, such as methyl, phenyl, vinyl, hydroxyl, or halogen. Adapted from [24].

**Figure 2 polymers-11-01685-f002:**
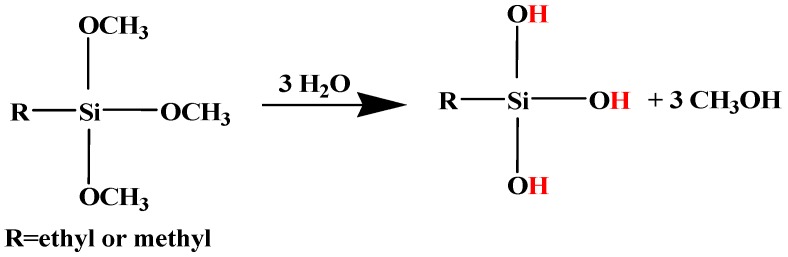
Representation of the general hydrolysis mechanism of silanes. Adapted from [30].

**Figure 3 polymers-11-01685-f003:**
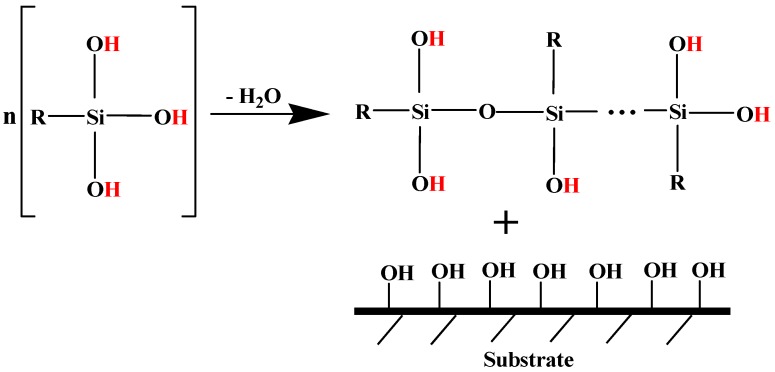
Representation of the typical condensation mechanism of silanes. Adapted from [30].

**Figure 4 polymers-11-01685-f004:**
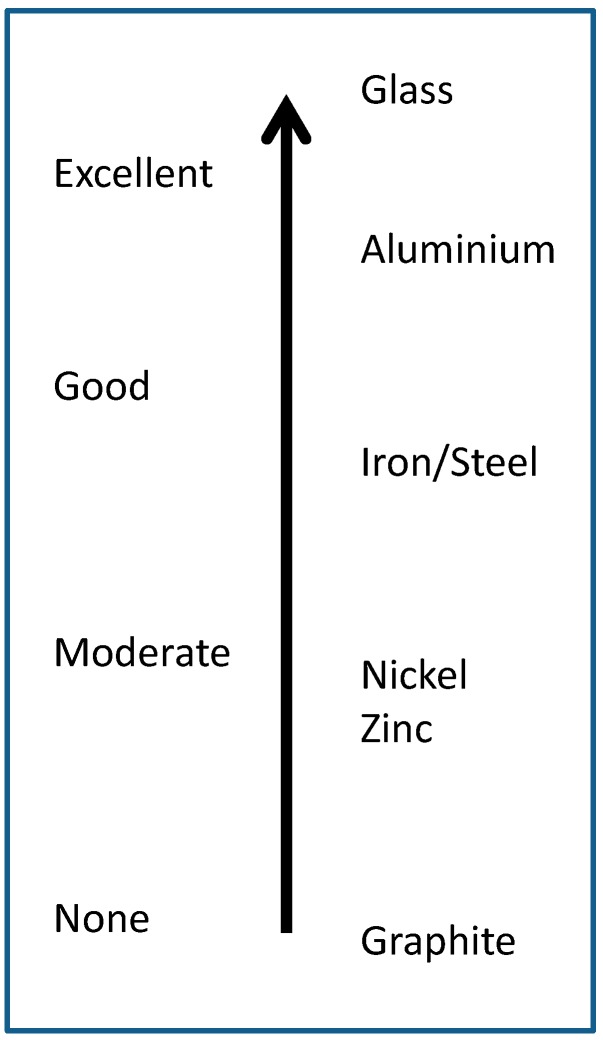
Schematic illustration of the effectiveness adhesion of silane-based adhesives and sealants (AS) on different substrates. Adapted from [24].

**Figure 5 polymers-11-01685-f005:**
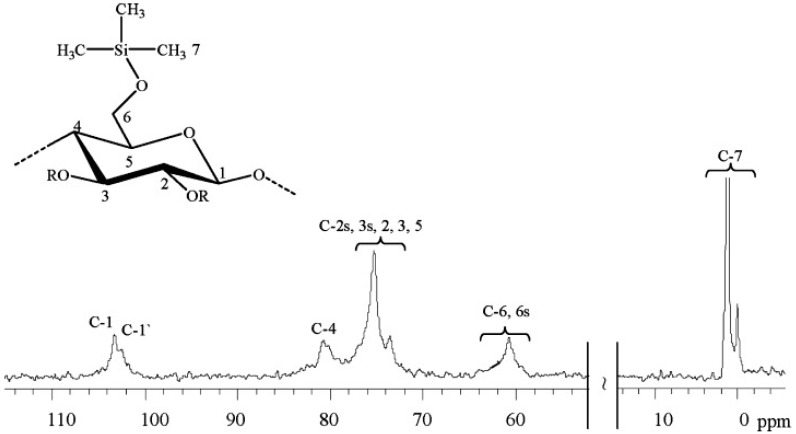
^13^C NMR spectrum of trimethylsilyl cellulose (degree of substitution, DS ≈ 0.43) in DMSO-d_6_, R means trimethylsilyl group or H according to DS. Reproduced from [45,46] with permission from Wiley and Copyright Clearance Center, 2008.

**Figure 6 polymers-11-01685-f006:**
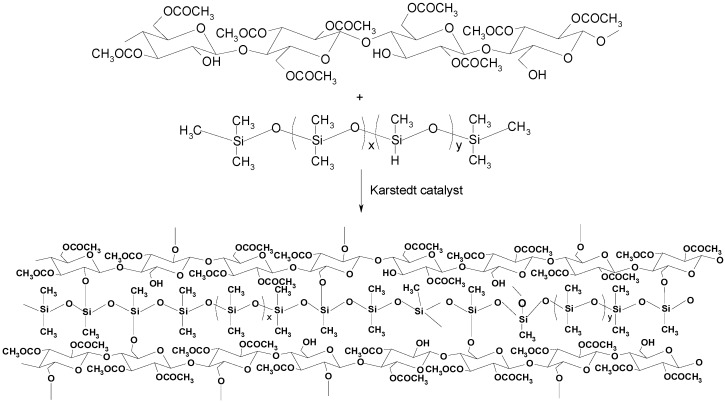
Cellulose silylation and the subsequent regeneration using acid vapor. Reproduced from [47] with permission from Springer and Copyright Clearance Center, 2010.

**Figure 7 polymers-11-01685-f007:**
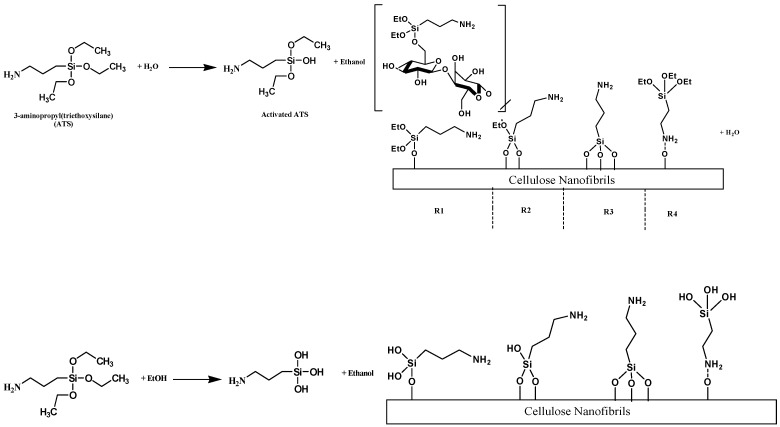
Schematic illustration of the reactions/interactions between aminopropyl triethoxysilane (ATS) and cellulose nanofibrils. Adapted from [49].

**Figure 8 polymers-11-01685-f008:**
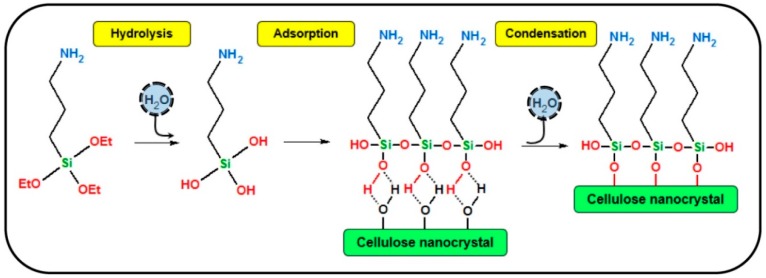
Reaction scheme between 3-aminopropyltriethoxysilane (APTES) and cellulose nanocrystals (CNC) in the presence of water. Reproduced from [53] with permission from Elsevier and Copyright Clearance Center, 2018.

**Figure 9 polymers-11-01685-f009:**
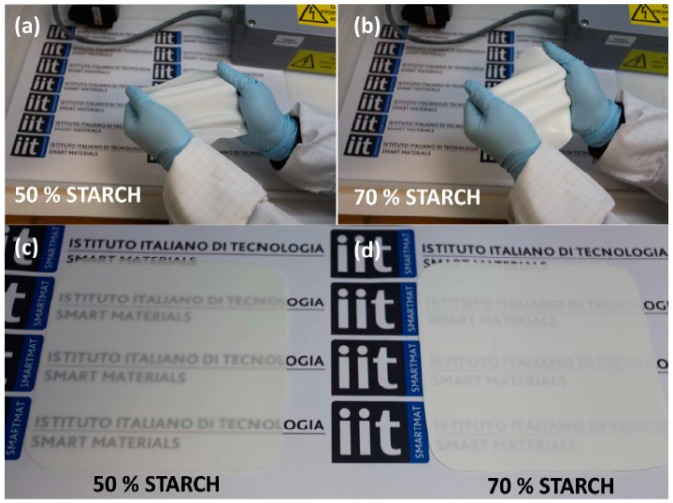
Photographs of bio-elastomers containing (**a**) 50 wt% and (**b**) 70 wt% of unmodified corn starch and examples of their transparency in (**c**,**d**), respectively. Adapted from [57] with permission from American Chemical Society and Copyright Clearance Center, 2015.

**Figure 10 polymers-11-01685-f010:**
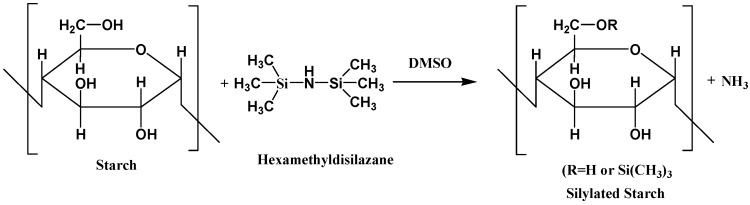
Schematic illustration of the reactions involved in the silylation of starch with hexamethyldisilazane (HMDS). Adapted from [58], with permission from Elsevier and Copyright Clearance Center, 2009.

**Figure 11 polymers-11-01685-f011:**
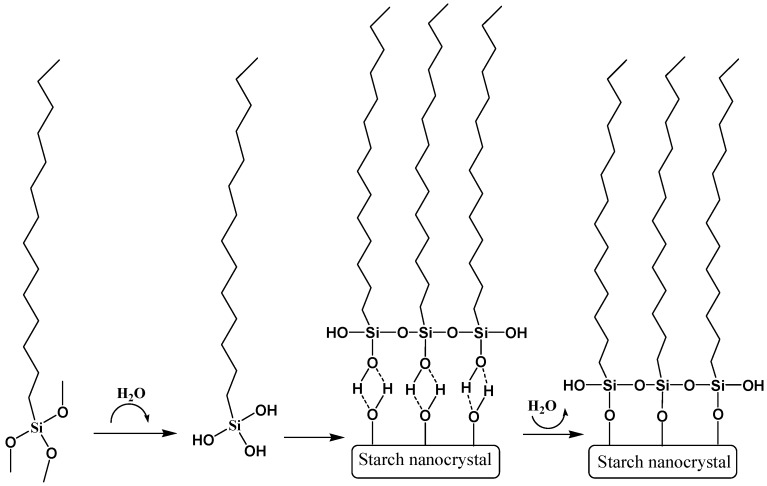
Reaction scheme between hexadecyltrimethoxysilane (HDS) and starch nanocrystals (SNC). Adapted from [60], with permission from Elsevier and Copyright Clearance Center, 2016.

**Figure 12 polymers-11-01685-f012:**
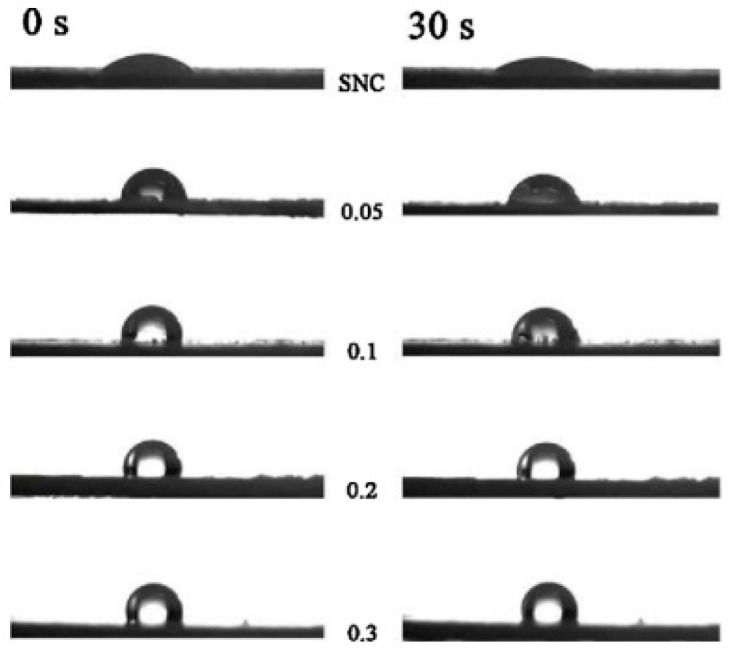
Photographs of a water drop on the surface of SNC and silylated SNC. Adapted from [60], with permission from Elsevier and Copyright Clearance Center, 2016.

**Figure 13 polymers-11-01685-f013:**
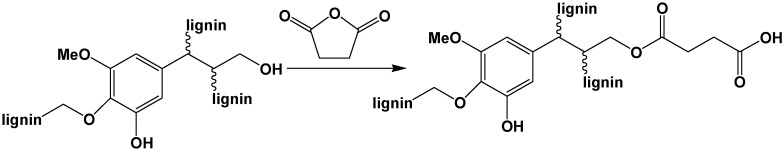
Schematic representation of the functionalization reaction between lignin and succinic anhydride (SAn) to produce succinylated lignin. Adapted from [65], with permission from American Chemical Society and Copyright Clearance Center, 2018.

**Figure 14 polymers-11-01685-f014:**
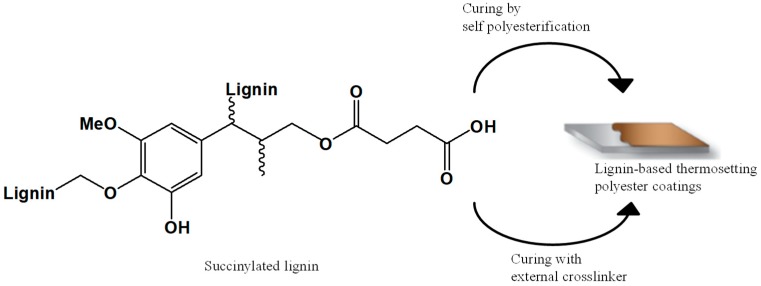
Process of curing the succinylated lignin in a substrate. Adapted from [65], with permission from American Chemical Society and Copyright Clearance Center, 2018.

**Figure 15 polymers-11-01685-f015:**
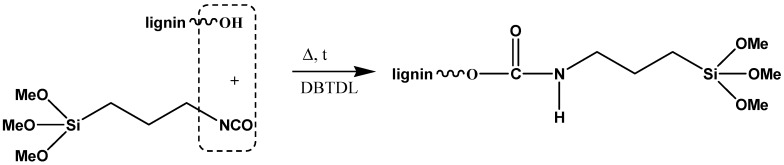
Schematic representation of the silanization of lignin. Adapted from [67], with permission from American Chemical Society and Copyright Clearance Center, 2019.

**Figure 16 polymers-11-01685-f016:**
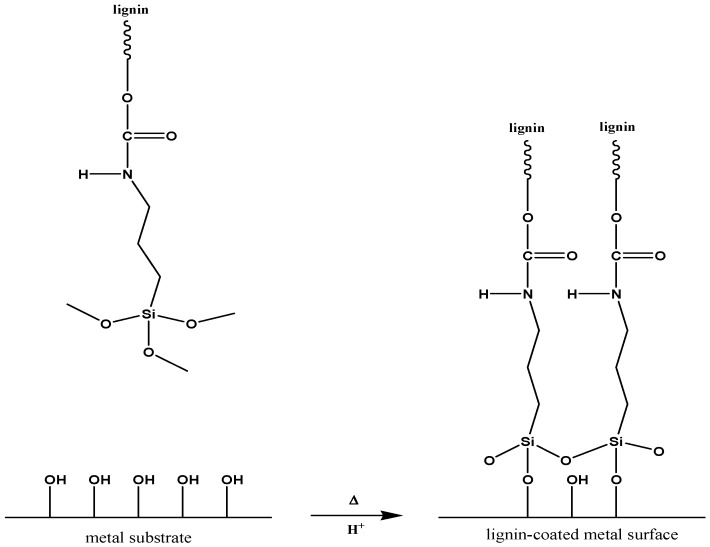
Schematic representation of the covalent bonding between lignin fractionalization and silanization (LF-S) and metal substrate promoted by the presence of acetic acid and heat. Adapted from [67], with permission from American Chemical Society and Copyright Clearance Center, 2019.

**Figure 17 polymers-11-01685-f017:**
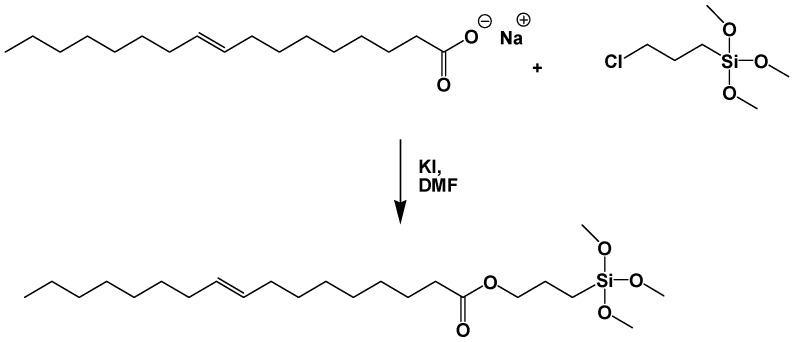
Scheme reaction of the synthesis of rapeseed-based silane. Adapted from [69], with permission from Springer Nature and Copyright Clearance Center, 2018.

**Figure 18 polymers-11-01685-f018:**
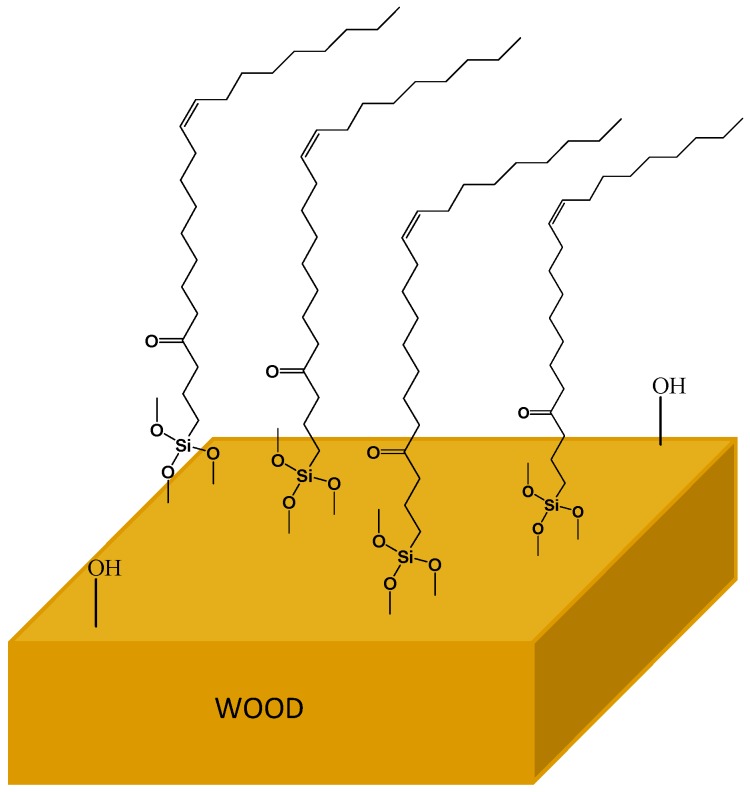
Schematic illustration of the covalent bonds formed between the silane groups of the vegetable oil-based coating and OH groups of wood. Adapted from [69], with permission from Springer Nature and Copyright Clearance Center, 2018.

**Figure 19 polymers-11-01685-f019:**
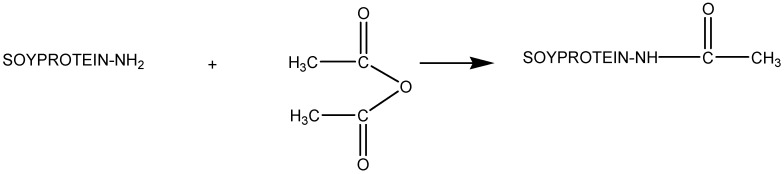
Acylation of an amino group of soy protein. Adapted from [79], with permission from Elsevier and Copyright Clearance Center, 2002.

**Figure 20 polymers-11-01685-f020:**
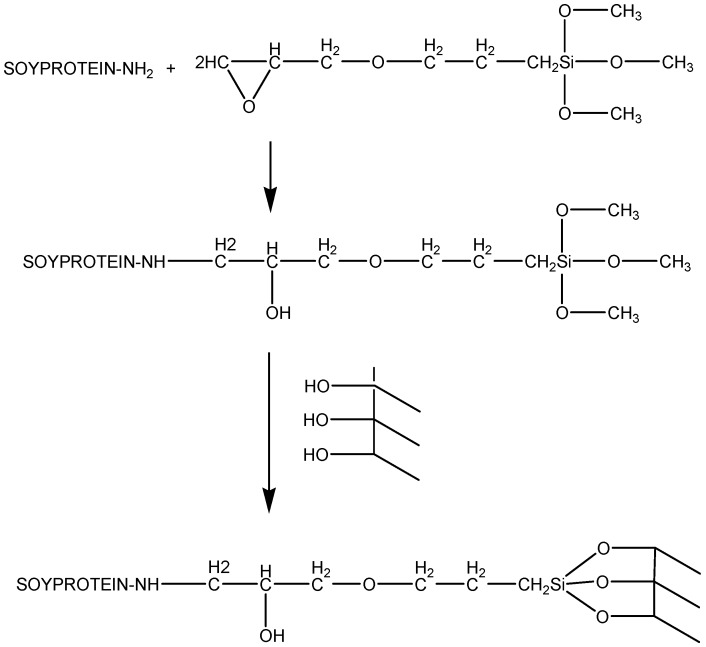
Silanation of soy protein for improving interfacial adhesion in glass fiber reinforced composites. Adapted from [79], with permission from Elsevier and Copyright Clearance Center, 2002.

**Table 1 polymers-11-01685-t001:** Thermal conductivity and electrical conductivity of the nanocomposites synthesized [54].

Samples	Electrical Conductivity (Scm^−1^)	Thermal Conductivity, k (Wm^−1^K^−1^)
PPy	5.422 × 10^−6^	0.343
TSA-doped PPy, pH 1	8.422 × 10^−3^	0.432
TSA-doped PPy, pH 3	1.407 × 10^−3^	0.422
TSA-doped PPy, pH 4	9.955 × 10^−4^	0.406
TSA-doped PPy:EC (80:20)	1.076 × 10^−3^	0.399
TSA-doped PPy:EC (70:30)	3.751 × 10^−5^	0.371
TSA-doped PPy:EC (50:50)	3.350 × 10^−8^	0.237

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
