# Peer review of "Brief Overview on Bio-Based Adhesives and Sealants"

_polymers, 2019, doi:10.3390/polym11101685_

Round 1

Reviewer 1 Report

The review by Solange et al provides a summary of different approaches in development of bio-based adhesive and sealants. Overall, the report provides a brief review of pertinent literature in bio-based adhesives & sealants for the readers particularly interested in this research space. However, the report lacks critical review of the available literature and authors hardly provide with their insights on recent developments/promises, problems/limitations, and future perspectives in this area. I recommend that the authors provide with critical reviews and their insights. In addition, the entire manuscript suffers from poor sentence structures and grammatical errors.

1.       Page 2, 2nd paragraph is one sentence paragraph, which is very unlikely in scientific writing. Please either elaborate this or combine this sentence with preceding/succeeding paragraphs.

2.       Extensive English polishing is necessary. Whereas some sentences are wordy/lengthy, some sentences are incomplete and hardly make sense. For instance, page 2, 3rd paragraph, the sentence “It is envisaged the decrease of manufacturing costs while creating a whole new “green” businesses opportunities to satisfy market needs”. It hardly makes sense. Similarly, page 3, 2nd paragraph, the sentence “On the other hand, the methyl groups, weakly interacting with each other, shield the main chain, as well as the length of the Si-O bond and the open angle there are contributing for giving the siloxane chain a high flexibility” is very wordy and does not clearly tell what information authors are trying deliver. Periods (commas) are used so frequently and unnecessarily that the sentences are very lengthy and complicated. This happens throughout the manuscript.

Some of my specific comments are:

3.       Page 2, 3rd paragraph, reference citation, please use same styling throughout the manuscript. For citation of some references, authors’ last name is provided whereas numbering is mainly used in the manuscript. Please be consistent.

4.       Section 2, 1st paragraph, maybe Figure 1?

5.       Page 3, 3rd paragraph is also one sentence paragraph, and is very wordy.

6.       Authors talk about triacetoxysilane, but the chemical structure presented in figure 2 is that of methyl trimethoxy silane. How easily the trimethoxy silane undergo hydrolysis with moisture? To the reviewer’s understanding, the ether linkages as shown in Figure 2 are relatively more stable than the acetoxy group talked in the manuscript.

7.       Some information is not relevant. For instance, the last paragraph of page 4 does not seem to be relevant in the scope of review.

8.       What is the intension of including figure 9 in the review as nothing about the figure has been described in the text?

9.       Page 13, 2nd last paragraph, authors mention “substituting part of the OH groups of starch by a bulky silyl group allows the reaction with (e)-caprolactone.” What is/was the purpose of this reaction (what product was formed)? And what were the properties and applications of the resulted product?

10.   Page 16, 1st paragraph, the figure number in the text should be 13

11.   Page 17, 2nd paragraph, the figure number in the text is 16?

Author Response

“The review by Solange et al provides a summary of different approaches in development of bio-based adhesive and sealants. Overall, the report provides a brief review of pertinent literature in bio-based adhesives & sealants for the readers particularly interested in this research space.”

Reply: We are thankful to the reviewer for recognizing the general importance of our manuscript.

“However, the report lacks critical review of the available literature and authors hardly provide with their insights on recent developments/promises, problems/limitations, and future perspectives in this area. I recommend that the authors provide with critical reviews and their insights. In addition, the entire manuscript suffers from poor sentence structures and grammatical errors.”

Reply: The manuscript has been changed in order to meet the reviewer concerns. Changes are properly highlighted in the revised text.

“1. Page 2, 2nd paragraph is one sentence paragraph, which is very unlikely in scientific writing. Please either elaborate this or combine this sentence with preceding/succeeding paragraphs.”

Reply: We agree with the reviewer suggestion and the sentence has been revised.

“2.Extensive English polishing is necessary. Whereas some sentences are wordy/lengthy, some sentences are incomplete and hardly make sense. For instance, page 2, 3rd paragraph, the sentence “It is envisaged the decrease of manufacturing costs while creating a whole new “green” businesses opportunities to satisfy market needs”. It hardly makes sense. Similarly, page 3, 2nd paragraph, the sentence “On the other hand, the methyl groups, weakly interacting with each other, shield the main chain, as well as the length of the Si-O bond and the open angle there are contributing for giving the siloxane chain a high flexibility” is very wordy and does not clearly tell what information authors are trying deliver. Periods (commas) are used so frequently and unnecessarily that the sentences are very lengthy and complicated. This happens throughout the manuscript.”

Reply: All the reviewer suggestions were taken into account. We have now carefully revised the entire text in order to minimize as much as possible the grammatical mistakes, typos and confusing sentences.

“3. Page 2, 3rd paragraph, reference citation, please use same styling throughout the manuscript. For citation of some references, authors’ last name is provided whereas numbering is mainly used in the manuscript. Please be consistent.”

Reply: We apologize for this odd mistake. The reference style has been corrected and made coherent along the entire manuscript.

“4. Section 2, 1st paragraph, maybe Figure 1?”

Reply: The figure number has been corrected.

“5.Page 3, 3rd paragraph is also one sentence paragraph, and is very wordy.”

Reply: The paragraph has been revised.

“6. Authors talk about triacetoxysilane, but the chemical structure presented in figure 2 is that of methyl trimethoxy silane. How easily the trimethoxy silane undergo hydrolysis with moisture? To the reviewer’s understanding, the ether linkages as shown in Figure 2 are relatively more stable than the acetoxy group talked in the manuscript.”

Reply: We do agree with the reviewer that the ether linkages are, in principle, more stable than the acetoxy groups. Nevertheless, it is out of the scope of this paper to compare the relative stability of all hypothetical silane-based molecules. Figure 2 is simply a rather generic and schematic representation of a hypothetical silicone molecule highlighting the typical chemical structure (backbone) and preferential substituting groups.

“7. Some information is not relevant. For instance, the last paragraph of page 4 does not seem to be relevant in the scope of review.”

Reply: Following the reviewer suggestion, the last paragraph of page 4 has been removed.

“8. What is the intension of including figure 9 in the review as nothing about the figure has been described in the text?”

Reply: We agree with the reviewer and a new sentence has been added in page 12 to highlight the mechanical robustness and transparency of the produced films presented in Figure 9.

“9. Page 13, 2nd last paragraph, authors mention “substituting part of the OH groups of starch by a bulky silyl group allows the reaction with (e)-caprolactone.” What is/was the purpose of this reaction (what product was formed)? And what were the properties and applications of the resulted product?”

Reply: One of the key requirements for the successful synthesis of poly-(e)-caprolactone (e-CL) grafted corn starch co-polymers, is the use of a homogeneous reaction medium for the ring-opening polymerization of starch with e-CL. A possible approach relies on making starch more hydrophobic by partial substitution of the OH groups by trimethylsilyl groups. The products may have interesting applications as compatibilizing agents for starch-polymer blends. Both the purpose of the silylation reaction and possible application of the final product were added to the revised manuscript.

“10. Page 16, 1st paragraph, the figure number in the text should be 13.”

Reply: The figure number has been corrected.

“11. Page 17, 2nd paragraph, the figure number in the text is 16?”

Reply: The figure number has been corrected.

Reviewer 2 Report

The review has been done thoroughly and well prepared. It is also articulated and worded very well and clearly. The matter of the review is up-to-date and prepared using a relatively large number of publications.

However, in a considerable number of figures no permission from the publisher and Copyright Clearance Center has been attained. Please ask for appropriate permission to use these Figures. There are also several other important publications that have been neglected in the list of publications cited in this review. Likewise, unfortunately there is also no any description of the tools used to reach the publications used in this review.

Below are some examples of important publications which I would expect to see citted in this review.

1. Recent Development of Biobased Epoxy Resins: A Review

Sudheer Kumar, Sushanta K. Samal, Smita Mohanty & Sanjay K. Nayak

Polymer-Plastics Technology and Engineering

Volume 57, 2018 - Issue 3

2. Evaluation of properties of starch-based adhesives and particleboard manufactured from them

Kushairi Mohd Salleh, Rokiah Hashim, Othman Sulaiman, Salim Hiziroglu, Wan Noor Aidawati Wan Nadhari, Norani Abd Karim, Nadiah Jumhuri & Lily Zuin Ping Ang,

Journal of Adhesion Science and Technology

Volume 29, 2015 - Issue 4

3. Soy-based adhesives for wood-bonding – a review

Doroteja Vnučec, Andreja Kutnar & Andreja Goršek

Journal

Journal of Adhesion Science and Technology

Volume 31, 2017 - Issue 8

Author Response

“The review has been done thoroughly and well prepared. It is also articulated and worded very well and clearly. The matter of the review is up-to-date and prepared using a relatively large number of publications.”

Reply: We acknowledge the reviewer for the very general very positive feedback.

“However, in a considerable number of figures no permission from the publisher and Copyright Clearance Center has been attained. Please ask for appropriate permission to use these Figures.”

Reply: Even though most of the figures have been redrawn and slightly changed by us we have decided to follow the reviewer suggestion and thus request the publisher permissions for the missing figures.

“There are also several other important publications that have been neglected in the list of publications cited in this review. Likewise, unfortunately there is also no any description of the tools used to reach the publications used in this review.

Below are some examples of important publications which I would expect to see cited in this review.

Recent Development of Biobased Epoxy Resins: A Review

Sudheer Kumar, Sushanta K. Samal, Smita Mohanty & Sanjay K. Nayak

Polymer-Plastics Technology and Engineering

Volume 57, 2018 - Issue 3

Evaluation of properties of starch-based adhesives and particleboard manufactured from them; Kushairi Mohd Salleh, Rokiah Hashim, Othman Sulaiman, Salim Hiziroglu, Wan Noor Aidawati Wan Nadhari, Norani Abd Karim, Nadiah Jumhuri & Lily Zuin Ping Ang,

Journal of Adhesion Science and Technology

Volume 29, 2015 – Issue 4

Soy-based adhesives for wood-bonding – a review

Doroteja Vnučec, Andreja Kutnar & Andreja Goršek

Journal

Journal of Adhesion Science and Technology

Volume 31, 2017 – Issue 8”

Reply: We thank the reviewer suggestions which have been included in the revised manuscript.  

Reviewer 3 Report

n my opinion, this review manuscript is poor organized. Many terms and concepts are used obscurely. The literature review seems not updated. It is hard to see any novel vision from this manuscript to shed light for future research in the reviewed field. Some exampled concerns are listed below.

Title is adhesives and sealants and so stated in Abstract and main text. However, the manuscript almost omit the field of wood adhesives (perhaps the most important adhesives in regards to the amount consumed). More or less, the authors should refer to several recent books and review articles.

In the introduction, the author should first give the definition of adhesives and sealants, their types etc, such as in Heinrich (2019).

The main text lacks substantial information on the critical parameters (numbers) of adhesives and sealants, such as adhesive strength, water resistance, and viscosity. The review is not worth reading without these true numbers (values) and how they are improved among those cited papers.

Do these organosilicon compounds really belong to bio-based products?

The very limited conclusion doest not fit the big title.

References suggested

He, Z., (ed.) 2017. Bio-based Wood Adhesives: Preparation, Characterization, and Testing, pp. 1-356. CRC Press, Boca Raton, FL.

Heinrich, L.A. 2019. Future opportunities for bio-based adhesives–advantages beyond renewability. Green Chemistry 21:1866-1888.

Petrič, M. 2019. Influence of Silicon‐Containing Compounds on Adhesives for and Adhesion to Wood and Lignocellulosic Materials: A Critical Review. Progress in Adhesion and Adhesives 4:25-76.

Pizzi, A., and K.L. Mittal, (eds.) 2011. Wood adhesives, pp. 1-451. CRC Press, Boca Raton, FL.

Author Response

“In my opinion, this review manuscript is poor organized. Many terms and concepts are used obscurely. The literature review seems not updated. It is hard to see any novel vision from this manuscript to shed light for future research in the reviewed field. Some exampled concerns are listed below.”

Reply: We hope the revised version solved most of the problems identified by the reviewer. In particular, the text and structure has been revised and some recent references have been included.

“Title is adhesives and sealants and so stated in Abstract and main text. However, the manuscript almost omit the field of wood adhesives (perhaps the most important adhesives in regards to the amount consumed). More or less, the authors should refer to several recent books and review articles.”

Reply: As the reviewer is aware the literature becomes enormous if we start widening the scope of the paper just a bit. Although we do recognize the vast importance of wood adhesives, it was not our intention to focus on a particular class/application of adhesives or sealants. We rather intended to briefly highlight the opportunities of using different bio-based raw materials, such as biopolymers, as friendly alternatives to non-renewable petroleum-based feedstock. Nevertheless, we do agree with the reviewer and the text has been revised and some more references have been added, including the ones suggested by the reviewer.

“In the introduction, the author should first give the definition of adhesives and sealants, their types etc, such as in Heinrich (2019).”

Reply: Indeed, this is a very important reference which has now been added.

“The main text lacks substantial information on the critical parameters (numbers) of adhesives and sealants, such as adhesive strength, water resistance, and viscosity. The review is not worth reading without these true numbers (values) and how they are improved among those cited papers.”

Reply: We do agree with the reviewer that these numbers are very important to properly characterize the different adhesives and sealants. Nevertheless, the different parameters are not always robust and comparable due to differences in “standard” methodology used, composition, conditions, etc. Therefore, we have decided to focus on the different bio-based sources, introducing some of the most relevant synthesis strategies adopted to develop novel systems, rather than to provide detailed and quantitative information on their physical-chemical features, performance or hypothetical functionality. Occasionally, this might be present but it was not our major concern.

“Do these organosilicon compounds really belong to bio-based products?”

Reply: Indeed, this is an important but tricky question. In all systems discussed, a bio-based raw material has been used and modified somehow with reactive silane units. Therefore, we believe that by replacing non-renewable petroleum-based feedstock by bio-based resources, such as biopolymers, we are not only improving the product sustainability but also developing systems with lower toxicity and higher biodegradability. Therefore, we are tempted to say that these bio-based organosilicon compounds can be considered bio-based products. Nevertheless, we are aware that this is a rather general and, perhaps, naïve perspective and thus a deep analysis for each system should be carefully performed in order to clearly answer this question.

“The very limited conclusion does not fit the big title.”

Reply: The conclusion has been revised.

“References suggested:

He, Z., (ed.) 2017. Bio-based Wood Adhesives: Preparation, Characterization, and Testing, pp. 1-356. CRC Press, Boca Raton, FL.

Heinrich, L.A. 2019. Future opportunities for bio-based adhesives–advantages beyond renewability. Green Chemistry 21:1866-1888.

Petrič, M. 2019. Influence of SiliconContaining Compounds on Adhesives for and Adhesion to Wood and Lignocellulosic Materials: A Critical Review. Progress in Adhesion and Adhesives 4:25-76.

Pizzi, A., and K.L. Mittal, (eds.) 2011. Wood adhesives, pp. 1-451. CRC Press, Boca Raton, FL.”

Reply: We thank the reviewer suggestions which have been included in the revised manuscript.

Reviewer 4 Report

Some of the abbreviations are not clearly defined.

The style of the figures is inconsistent (e.g., font).

Page 3, Line 3: Figure 1?

Page 5, Lines 26 and 28: Please use either “gamma LG” or “gamma L.”

Page 6, Line 4: Figure 4?

Page 6, Line 19: Please remove the clause “Although the literature is surprisingly scarce in this area.”

Page 8, Line 13: Please clarify Klemm et al.

Page 11, Line 9: Please clarify Draman et al.

Page 13, Line 4: Please clarify Petzold et al.

Page 13, Line 15: Please clarify Wei et al.

Author Response

“Some of the abbreviations are not clearly defined.”

Reply: Abbreviations were reviewed and modified when necessary.

“The style of the figures is inconsistent (e.g., font).”

Reply: The font of figures has been revised.

“Page 3, Line 3: Figure 1?”

Reply: The figure number has been corrected.

“Page 5, Lines 26 and 28: Please use either “gamma LG” or “gamma L.”

Reply: The typo has been corrected.

“Page 6, Line 4: Figure 4?”

Reply: The figure number has been corrected.

“Page 6, Line 19: Please remove the clause “Although the literature is surprisingly scarce in this area.”

Reply: As suggested by the reviewer, the indicated sentence has been removed.

“Page 8, Line 13: Please clarify Klemm et al.

Page 11, Line 9: Please clarify Draman et al.

Page 13, Line 4: Please clarify Petzold et al.

Page 13, Line 15: Please clarify Wei et al.”

Reply: References have been clarified by adding the proper number.

Round 2

Reviewer 1 Report

The revised manuscript has been significantly improved and reads better, but still suffers from several grammatical errors and poor sentence structure. A thorough revision of sentence structure, grammatical errors, and typos is necessary.

Author Response

We are thankful to the reviewer for recognizing our efforts in improving the manuscript while addressing all the reviewer’s comments and questions. The manuscript has now been extensively revised by different native English speakers and we believe most of the issues raised by the reviewer have been identified and corrected (main changes highlighted in yellow).

Reviewer 3 Report

The revised version reads much better. With the limited information in the manuscript, the title with the term "overview" is appareantly overstressed. Indeed, the authors themselves claimed it as a brief overview in Abstract (line24). Thus, suggest to change the title as " Mini View on Bio-based Adhesives and Sealants".

Author Response

We thank the reviewer for recognizing the improvements done in the manuscript. We also agree the title might sound too ambitious and therefore we have changed it to “Brief overview on bio-based adhesives and sealants”.
